# The Effectiveness of Whole-Body Vibration and Heat Therapy on the Muscle Strength, Flexibility, and Balance Abilities of Elderly Groups

**DOI:** 10.3390/ijerph20021650

**Published:** 2023-01-16

**Authors:** Shiuan-Yu Tseng, Chung-Liang Lai, Chung-Po Ko, Yu-Kang Chang, Hueng-Chuen Fan, Chun-Hou Wang

**Affiliations:** 1Department of Senior Services Industry Management, Minghsin University of Science and Technology, Hsinchu 30401, Taiwan; 2Department of Occupational Therapy, Asia University, Taichung 41354, Taiwan; 3Department of Physical Medicine and Rehabilitation, Puzi Hospital, Ministry of Health and Welfare, Chiayi 61347, Taiwan; 4Department of Neurosurgery, Tungs’ Taichung MetroHarbor Hospital, Taichung 43503, Taiwan; 5Department of Post-Baccalaureate Medicine, College of Medicine, National Chung Hsing University, Taichung 40227, Taiwan; 6Department of Medical Research, Tungs’ Taichung MetroHarbor Hospital, Taichung 43503, Taiwan; 7Nursing and Management, Jenteh Junior College of Medicine, Miaoli 35664, Taiwan; 8Department of Pediatrics, Tungs’ Taichung MetroHarbor Hospital, Taichung 43503, Taiwan; 9Department of Physical Therapy, Chung Shan Medical University, Taichung 40201, Taiwan; 10Physical Therapy Room, Chung Shan Medical University Hospital, Taichung 40201, Taiwan

**Keywords:** whole-body vibration, heating pad, middle-aged and older individuals, balance training, flexibility training

## Abstract

Whole-body vibration (WBV) is a novel exercise training measure that promotes the muscle strength, flexibility, and balance abilities of elderly groups. The feasibility and applicability of 20–30 min (lowering a heat pack at 73 °C by wrapping it in multiple layers of towels to 40–43 °C before it touched the skin) thermotherapy are increasingly being demonstrated by applications and clinical trials. Studies show that it increases the flexibility of macules and ligament. However, no studies have examined the interactions between the pre-exercise and post-exercise application of heat therapy (duration a training course). Therefore, this study investigates the effects of WBV and heat therapy on the muscle strength, flexibility, and balance abilities of elderly groups. Eighty middle-age and elderly participants with no regular exercise habits were enrolled in this study. They were randomly assigned to a WBV group, a WBV plus heat therapy group, a heat therapy alone group, and a control group. The WBV groups underwent 5-min, fixed-amplitude (4 mm), thrice-weekly WBV training sessions for 3 consecutive months on a WBV training machine. Participants’ balance was measured using the limits of stability (LOS) test on a balance system. The pretest and posttest knee extensor and flexor strength were tested using an isokinetic lower extremity dynamometer. Pretest and posttest flexibility changes were measured using the sit-and-reach test. Significantly larger pretest and posttest differences in flexibility and muscle strength were observed in the WBV and WBV plus heat therapy groups. The addition of heat therapy to WBV resulted in the largest flexibility improvements.

## 1. Introduction

The Population Reference Bureau (PRB) estimated that individuals older than 65 accounted for 8% of the world’s 7.06 billion population in 2012, with 35% of these individuals living in higher-income regions. One of four or five individuals in Japan, Monaco, Germany, and Italy belongs to this demographic cohort, suggesting that population aging is a worldwide phenomenon [1]. Individuals have longer lifespans in today’s aging society. Globally, the proportion of individuals older than 60 years is expected to be twice the current figure by 2050 [2,3]. Based on American Centers for Disease Control and Prevention (CDC) 2010 data, the annual rate of falls among individuals older than 65 is 30% (at least one fall per year), whereas the annual rate among those over 80 years of age exceeds 50% [4].

Balance impairment has been identified as a salient fall risk. Reduced postural stability is associated with age-related deficits in sensory function, central processing, and neural pathways for musculoskeletal strength and motor control [5]. A wide body of research has demonstrated that exercise enhances the functional performance of elderly groups. Many exercise interventions and therapies for postural control enhancement in elderly groups have been directed toward promoting muscle strength, flexibility, aerobic capacity [6,7], visual feedback training [8], aerobic dancing [9], and exercise ball training [10]. However, studies show that nearly half of people over the age of 60 are physically inactive [11]. Therefore, how to effectively motivate the elderly adapt to exercise and choose the right exercise becomes critical. Several studies reported no positive effects of strength training on balance abilities [12,13,14]. Such differences may be related to the important vibration training parameter (frequency) [15]. However, Messier et al.’s (2000) [16] long-term study suggested that strength training enhances postural stability [17,18]. Therefore, regular and effective exercise and training methods are necessary for improving the balance abilities of elderly groups [8,19].

Whole-body vibration (WBV) training is a novel exercise method that takes place on a vibrating platform that generates continuous sinusoidal vibrations transmitted from the feet to the entire body to stimulate receptors in the body and create modulating effects. Vibratory stimuli were first documented to reduce the tone of spastic muscles [20]. Subsequent studies in the 1990s found that vibratory stimuli can help competitive athletes gain power and enhance sports performance during training [20]. In recent years, numerous animal and clinical studies have demonstrated the positive effects of WBV exercise on bone density and blood circulation in the lower extremities [21,22]. With regard to neuromuscular performance, studies have shown that WBV exercise enhances the muscle strength [23], flexibility [24], and sports performance [25] of athletes and typical adults alike. Additionally, WBV functions as a quick way of warming up before an exercise or training session and reduces pain caused by delayed onset muscle soreness (DOMS) [26]. Numerous studies have revealed the positive effects of WBV exercises on the balance abilities and functional performance of elderly groups [17,18], in addition to being a safe and convenient form of exercise [27]. WBV has been included in exercise therapies for stroke and multiple sclerosis patients [28,29].

Heat therapy is closely associated with exercise [30] due to the physiological benefits it confers on tissue metabolism, circulation, neuromuscular performance, and connective tissue functions [31]. Heat increases metabolic rates, vasodilation and blood flow, muscle relaxation, and connective tissue elasticity, thus promoting tissue recovery and providing pain relief by increasing the pain threshold [32,33]. According to literature, heat therapy induces muscle hypertrophy [34] and increases sports performance [35]. A recent study reported a high correlation between work output and muscle temperature. Based on the type and speed of muscle contraction, every 1 °C increase in muscle temperature increases subsequent sports performance by 2% to 5% [36]. Moreover, there is a positive correlation between muscle temperature changes and exercise speed [36]. To date, most studies on heat therapy focus on its effects on muscle strength and high-intensity exercise. Few studies have examined its effects on the important variable of the muscle endurance of elderly groups [36]. Moreover, there is a lack of relevant studies on the effects of long-term WBV training in combination with heat therapy on middle-aged and elderly groups. For these reasons, this study sought to uncover the effects of WBV and heating pads on the muscle strength, balance abilities, and flexibility of elderly groups. The results of this study can serve as a basis for subsequent studies on the correlation between WBV and heat therapy.

## 2. Materials and Methods

This study employed the single-blind randomized controlled trial design to prevent the examiners from knowing if a participant belonged to the experimental or control group. This study was approved by the hospital’s clinical research ethics committee (Ministry of Health and Welfare Tsaotun Psychiatric Center Institutional Review Board IRB104056, Clinical Trial Registration Number ChiCTR-IOR-16008059). All participants were fully informed of the contents of this study beforehand and had signed a consent form.

This study enrolled 80 middle-aged and elderly individuals. The inclusion criteria are as follows: (1) Individuals older than 45 years who lack regular exercise habits or have not received exercise instruction. According to the latest Taiwan Social Change Survey (TSCS), participant who answered that they exercised “several times a week” or “every day” were excluded from the study [37] because they exhibited exercise habits. (2) Are able to stand on their own without any assistance. The exclusion criteria are as follows: (1) Are taking drugs that interfere with musculoskeletal metabolism; (2) Had experienced bone fractures or underwent surgery within the past three months; (3) Have a history of dizziness.

Intervention conditions: Eligible participants were randomly assigned to three experimental groups and a control group. The vibration instrument used in this study was a commercial-grade WBV training machine (LV-1000, X-trend, Taiwan). Participants were randomly assigned to a WBV group (20 Hz, *n* = 20), a WBV plus heat therapy group (20 Hz + heat, *n* = 22), a heat therapy alone group (*n* = 21), and a control group (*n* = 17). The mode of vibration was fixed sinusoidal waves (amplitude: 0–4 mm). Participants trained for five minutes while standing in a natural position. Participants who underwent WBV stood on the training machine thrice weekly for three consecutive months (see Figure 1). A thermal group was added after vibration training was performed. Those who underwent heat therapy rested the back of their lower extremities (hamstrings and gastrocnemii) on a circulating water heating pad maintained at 40 °C for 20 min (CW89 WiPOS, Taiwan) (Figure 2) [38,39]. The control group was used only for measurements and to monitor exercise habits.

Outcome measure: Before undertaking the formal test, the participants first engaged in two familiarization exercises (for 10 s per session). The test parameters include each participant’s pre-WBV training, 3 months post-WBV training, and 6 months post-WBV training balance abilities and muscle strength, as well as the 1-year incidence of falls for all participants. Participants’ balance abilities were measured using the limits of stability (LOS) test on a balance system (Biodex Balance System, Shirley, NY, USA). The LOS test mainly challenges an individual’s center of gravity and movements within their base of support. In each test, participants are repeatedly instructed to shift their center of gravity to a randomly designated position according to the machine’s instructions and then return to the starting position afterwards. Additionally, this index (LOS) assesses participants’ ability to control their center of gravity in different directions.

Data processing: Before undertaking this test, a participant’s age, height, and stability grade were input into the balance testing machine. The test results, expressed in percentage, were derived from the ratio of the actual displacement of a participant’s center of pressure under the feet to the vertical distance between the center of pressure and the target position. The score is calculated as follows: (vertical distance to the target position/sample size) × 100% This balance ability assessment system has been demonstrated to have a good validity (ICC: 0.60–0.95) [40,41].

Muscle strength was measured using an isokinetic lower extremity dynamometer (Biodex System IV Pro, Shirley, NY, USA). The pretest and posttest knee extensor and flexor strength were tested along with their isokinetic concentric and eccentric muscle strength at a 60°/s angular speed.

After warming up, the participants were instructed to sit on the Biodex isokinetic dynamometer. Their trunk, pelvis, lower right thigh, and ankles were secured on the seat according to the instructions on the machine to prevent them from shifting their position during the rest and affecting the test results. The relative position of the seat and the dynamometer were adjusted such that the knee joints’ centers of rotation (lateral malleoli of the tibia) are aligned with the dynamometer’s axis of rotation. Isokinetic flexor strength tests began after the calf center of gravity was corrected. The isokinetic concentric contraction mode at a 60°/s angular speed was applied during the tests. Each participant underwent three tests, with a 1-min rest between each contraction.

Flexibility was measured through the sit-and-reach test, which assesses flexibility of the waist and lower limb joints. Participants are first instructed to sit on the floor or a mat with their shoes off, their thighs at shoulder width, the knee joints extended, and the toes facing upwards. A tape measure is placed between the legs. The bottom of the heel must be aligned with the 25-cm mark on the tape measure. The hands are placed on top of one another so that the middle fingers align, palms together. The participant then reaches to touch the tape measure for two seconds using their middle finger, and the farthest distance (in cm) is recorded. During the tests, the first reading is considered a practice result and the better of the latter two readings is taken as the final result [42].

The treatment rate for falls was assessed through phone interviews and pen-and-paper questionnaires about the participants’ history of being treated for falls within the last year.

The participants’ basic information and measurement variables were recorded as descriptive statistics. Chi-squared tests and one-way ANOVA were used to examine for differences between the four groups, while repeated-measure ANOVA was used to analyze the pretest and posttest differences of the descriptive statistics across all groups and the pretest and posttest differences in their LOS, muscle strength and flexibility. Data analysis was performed using SPSS 14.0 statistical software. The level of significance (α) was at 0.05 in all tests. Post-hoc analyses were performed through the Scheffe test in the case of significant between-group and pretest/posttest interactions.

## 3. Results

As shown in Figure 3, the study sample comprised 80 middle-aged and elderly participants. The participants were randomly assigned to a WBV training group (20 Hz, *n* = 20), a WBV training plus heat therapy group (20 Hz + heat, *n* = 22), a heat therapy alone group (*n* = 21), and a control group (*n* = 17).

As shown in Table 1, no significant differences were observed in the participants’ pretest indicators including age, height, weight, BMI, flexibility, muscle strength, and balance abilities (*p* > 0.05).

In terms of flexibility, significant pretest and posttest differences in flexibility were observed in the WBV and WBV plus heat therapy groups. The latter’s degree of improvement was significantly higher than that of the other three groups in three months (*p* = 0.007) (see Figure 4; Table 2).

In terms of muscle strength, no significant differences in muscle strength were observed in the heat therapy alone and control groups. A significant improvement in muscle strength was observed in the WBV and WBV plus heat therapy groups. The latter’s degree of improvement was significantly higher than those of the other three groups (see Figure 5; Table 3).

In terms of balancing abilities, no significant differences in balance abilities were observed in the heat therapy alone and control groups. And our study finding significant improvements in balance abilities were observed in the WBV and WBV plus heat therapy groups (see Figure 6; Table 4).

## 4. Discussion

Significant pretest and posttest differences in flexibility were observed in the WBV and WBV plus heat therapy groups. In other words, both the WBV groups improved their flexibility, muscle strength, and balanced functional performance. One theory suggests that vibration exercise promotes the neurotransmission characteristics of γ-aminobutyric acid as well as muscle spindle sensitivity, which stimulates more motor neurons [43]. The WBV plus heat therapy group had significantly more improvement than the other three groups. This finding is consistent with previous studies. WBV plus heat therapy groups were significantly higher than WBV group (especially in flexibility and muscle strength and the training effect remains only in the WBV plus heat therapy groups). This finding agrees with previous studies. There have been previous studies that suggest low-intensity resistance training combined with heat stress leads to increased muscle strength and volume, possibly due to the expression of heat shock protein by heat stress, and that heat stress promotes the hypertrophy of muscles induced by mechanical stress [30]. WBV exercise has been demonstrated to improve intramuscular temperature and a warming technique [24]. It has also been proven to enhance flexibility [44]. The high degree of improvement achieved by the WBV plus heat therapy group may be associated with the common application of heat therapy as an assistive pre-exercise measure [45]. This measure improves tissue metabolism and subsequently promotes pre-exercise muscle metabolism [46]. Heat also promotes glycogenesis and muscle recovery [46]. Another study found that heat therapy reduces the chance of tissue damage by promoting tissue flexibility and lowers the energy cost of muscle contraction by reducing internal friction [47].

No significant differences in muscle strength were observed in the heat therapy alone and control groups. This finding does not align with previous studies [34], which may be attributed to the different ages of the participants [48]. However, some studies have pointed out that active warming up induces more metabolic changes than passive warming up, thus increasing an individual’s preparedness for subsequent exercise tasks [36].

The WBV and WBV plus heat therapy groups significantly improved their muscle strength. This finding agrees with previous studies (41 °C) (41 °C, 60 min) [49] that found WBV could enhance muscle power [50,51]. Like previous studies, the degree of improvement achieved by the WBV plus heat therapy group was significantly higher than those of the other three groups, which could be due to the activation of heat-shock proteins, regarded as a factor contributing to improvements in resistance training outcomes [52]. High temperatures accelerate muscle glycogen breakdown. The passive increase of intramuscular temperature is associated with faster adenosine triphosphate (ATP) reactions, which are achieved by increasing the hydrogen ion consumption rate of creatinine phosphate (PCr) and promoting anaerobic glycolysis and muscle glycogen breakdown [53]. Another possible reason could be because heat is used in different forms before exercise [54]. In general, the temperatures of the muscles and ligaments are below that of the core temperature [54], which is around 37 °C in humans [55], while the skin temperature in the arms and legs is around 31 °C [56]. The temperatures of subcutaneous tissues differ by their depth and position relative to the core. Tissues located deeper and further away from the core often have lower temperatures [57]. Therefore, it is important to warm these regions before exercise. No significant differences in balance abilities were observed in the heat therapy alone and control groups after study started Six months. It has been shown in previous studies that if training stops for more than three to eight weeks, the arterial oxygen difference decreases. The rapid and progressive decrease in oxidase activity leads to a decrease in mitochondrial ATP production. These changes are associated with the decrease in VO_2max_ observed during prolonged cessation of training. Non-athletes with short-term training usually return to their baseline values after a short period of time without exercise [58]. It is therefore crucial to maintain long-term exercise training.

The WBV and WBV plus heat therapy groups significantly improved with balance abilities. This finding agrees with previous studies [51,58]. The general belief is that improved muscle strength also increases control of one’s balance abilities [51,58]. Many reports on the effectiveness of WBV for improving joint range and sensitivity have proven the active benefits of WBV training on dynamic stability [58]. No significant improvements in balance abilities were observed in the WBV and WBV plus heat therapy groups after six months. Studies indicate that after a month of inactivity, the adaptive performance of the muscles may return to the beginning state [59]. Therefore, we must consider how to maintain the effect of our exercise trainings. Future studies can broaden the age groups of participants or enroll athletes from different sports to participate in WBV training.

This study features the first major study to examine the combination of WBV plus heat training, and found that heat does indeed enhance the effect of exercise. The limitation of this study is that it is challenging to encourage non-exercising individuals to maintain exercise for three months, hence, the sample size remained small.

## 5. Conclusions

This study demonstrates that the WBV and addition of heat therapy in WBV training can improve muscle strength and flexibility. To increase flexibility, add a hot compress for more effective results. This treatment can be applied in WBV training to increase the aforementioned parameters. And if you want to maintain long-term exercise effects, it is recommended to continue to exercise to maintain good muscle strength performance.

## Figures and Tables

**Figure 1 ijerph-20-01650-f001:**
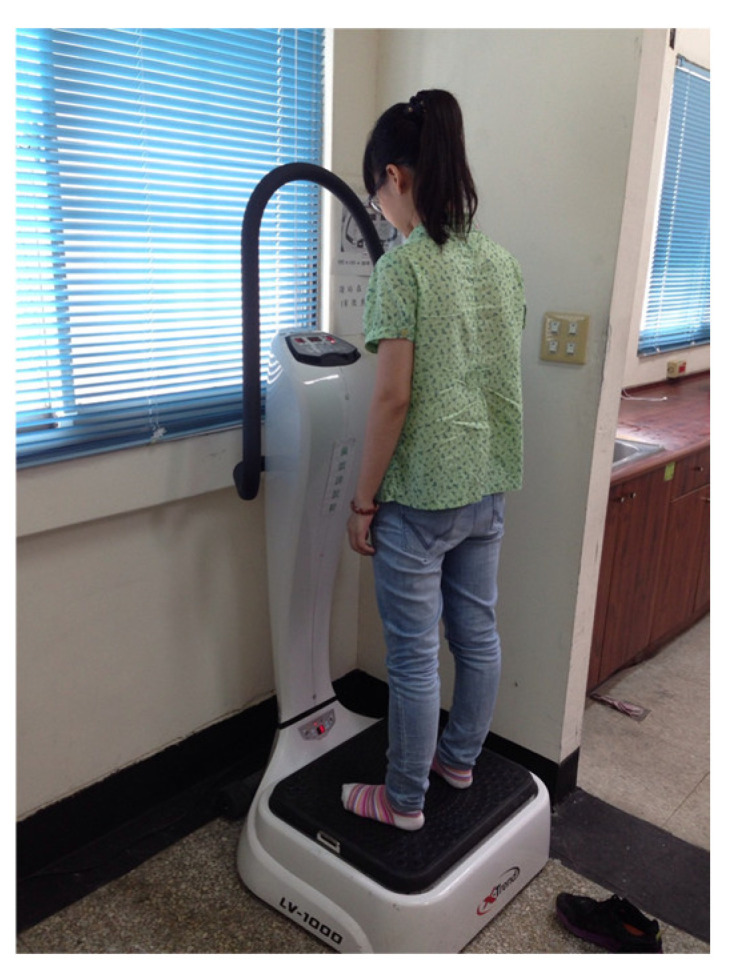
A participant standing on the WBV machine.

**Figure 2 ijerph-20-01650-f002:**
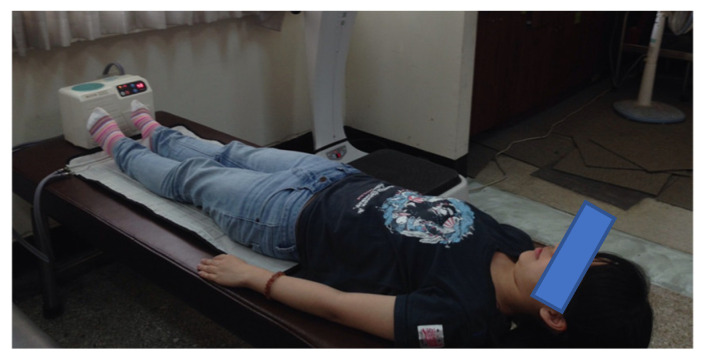
A heating pad warming a participant’s lower extremities.

**Figure 3 ijerph-20-01650-f003:**
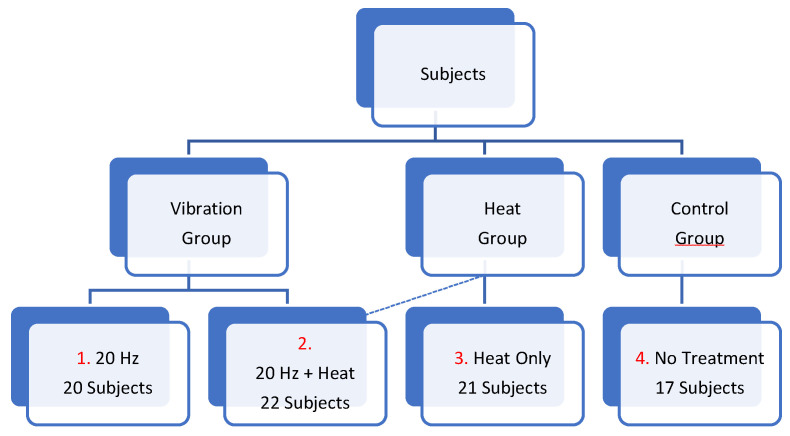
Participant group assignment.

**Figure 4 ijerph-20-01650-f004:**
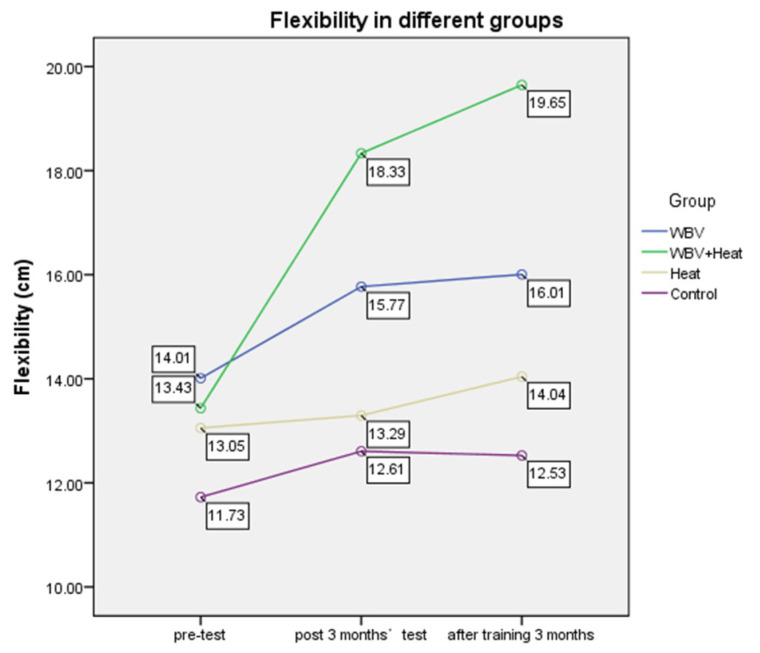
Flexibility in six months with different groups.

**Figure 5 ijerph-20-01650-f005:**
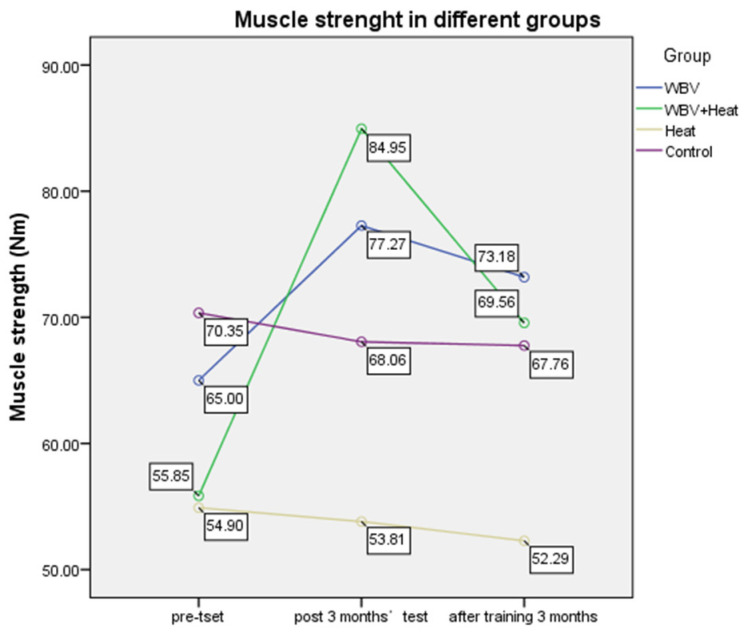
Muscle strength in six months with different groups.

**Figure 6 ijerph-20-01650-f006:**
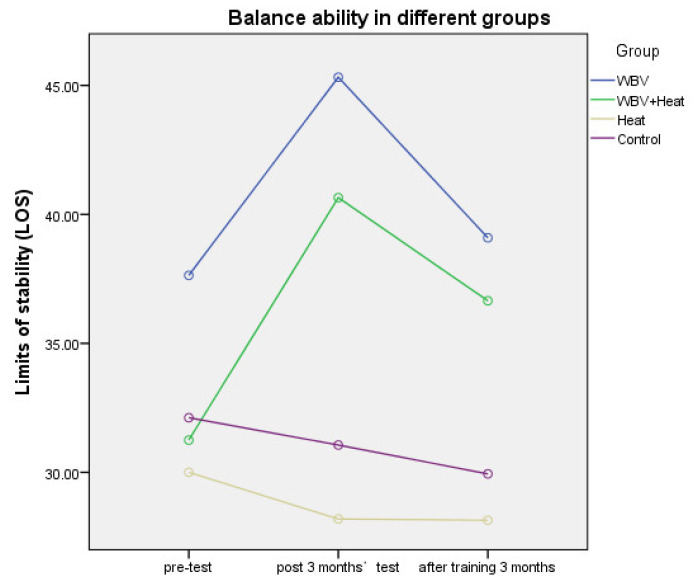
Balance in Six months with different groups.

**Table 1 ijerph-20-01650-t001:** ANOVA results.

		Sum of Squares	df	Mean Square	F	Significance
Age	Between groups	31.015	3	10.338	0.110	0.954
	Within groups	7110.872	76	93.564		
	Total	7141.888	79			
Height	Between groups	162.125	3	54.042	0.956	0.418
	Within groups	4294.074	76	56.501		
	Total	4456.199	79			
Weight	Between groups	79.823	3	26.608	0.249	0.862
	Within groups	8108.659	76	106.693		
	Total	8188.482	79			
BMI	Between groups	1.356	3	0.452	0.044	0.988
	Within groups	777.319	76	10.228		
	Total	778.676	79			
Pre-test flexibility	Between groups	52.464	3	17.488	0.299	0.826
	Within groups	4446.933	76	58.512		
	Total	4499.397	79			
Pre-test muscle strength	Between groups	3146.558	3	1048.853	1.079	0.363
	Within groups	73,854.242	76	971.766		
	Total	77,000.800	79			
Pre-test LOS	Between groups	734.882	3	244.961	1.509	0.219
	Within groups	12,334.606	76	162.297		
	Total	13,069.488	79			

**Table 2 ijerph-20-01650-t002:** Comparison of the Training Effect on Flexibility in Six Months.

Group	1. before Training	2. after Training for 3 Months	3. after Training for 6 Months	Time 1 vs. 2 *p*; Time 1 vs. 3 *p*
WBV	14.0 ± 8.92	15.7 ± 8.81	16.0 ± 8.74	0.024 *; 0.015 *
WBV + Heat	13.4 ± 7.08	18.3 ± 6.84	19.6 ± 10.11	0.000 *; 0.001 *
Heat	12.0 ± 6.89	13.2 ± 5.70	14.0 ± 6.28	0.877; 0.536
Control	11.7 ± 7.36	12.6 ± 8.21	12.5 ± 7.25	0.398; 0.389

A significant effect of time on flexibility performance was observed (*p* = 0.000) as well as an interaction between groups and time (*p* = 0.009). In the posthoc comparison, the WBV + Heat group was significantly better than the control group (*p* = 0.046). * *p* < 0.05; the values are given as mean (95% confidence interval).

**Table 3 ijerph-20-01650-t003:** Comparison of the Training Effect on Muscle Strength in Six Months.

Group	1. before Training	2. after Training for 3 Months	3. after Training for 6 Months	Time 1 vs. 2 *p*; Time 1 vs. 3 *p*
WBV	65.0 ± 28.93	77.2 ± 37.24	73.1 ± 35.75	0.013 *; 0.066
WBV + Heat	55.8 ± 32.30	84.9 ± 36.22	69.5 ± 35.13	0.000 *; 0.000 *
Heat	54.9 ± 32.02	53.8 ± 34.14	52.2 ± 31.72	0.723; 0.456
Control	70.3 ± 31.55	68.0 ± 31.45	67.7 ± 32.26	0.510; 0.463

The factor of time had a significant effect on muscle strength (*p* = 0.000) and there was also an interaction between different groups and time (*p* = 0.000). * *p* < 0.05; the values are given as mean (95% confidence interval).

**Table 4 ijerph-20-01650-t004:** Comparison of the Training Effect on Balance Performance in Six Months.

Group	1. before Training	2. after Training for 3 Months	3. after Training for 6 Months	Time 1 vs. 2 *p*; Time 1 vs 3. *p*
WBV	37.6 ± 11.24	45.3 ± 13.98	39.0 ± 12.63	0.010 *; 0.068
WBV + Heat	31.2 ± 11.87	40.6 ± 13.96	36.6 ± 15.06	0.011 *; 0.193
Heat	30.0 ± 15.80	28.1 ± 11.76	28.1 ± 11.32	0.480; 0.480
Control	32.1 ± 9.77	31.0 ± 16.84	29.9 ± 13.96	0.715; 0.715

In balance performance, time had a significant effect on equilibrium (*p* = 0.016), and there was also an interaction between different groups and time (*p* = 0.009). * *p* < 0.05; the values are given as mean (95% confidence interval).

## Data Availability

The data presented in this study are included in this published article.

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
