# Peer review of "The Effectiveness of Whole-Body Vibration and Heat Therapy on the Muscle Strength, Flexibility, and Balance Abilities of Elderly Groups"

_ijerph, 2023, doi:10.3390/ijerph20021650_

Round 1

Reviewer 1 Report

This is a very interesting study examining the effects of whole-body vibration and heat therapy for physical function. The design and outcome measures appear appropriate, but there are key parts of the results that have not been adequately reported. The discussion will also benefit from additional content after the authors have addressed the missing elements in the results section.

Abstract

- Please add a brief sentence for what the heat therapy involved, just like how you described the WBV condition.

- Please clarify what pre-test and post-test are in reference to: each exercise session? Or the whole 3-month block?

- Keywords: you could add more keywords to make your paper more searchable, e.g. exercise, strength training, flexibility training. Consider using MeSH terms.

1. Introduction

Line 38-39: annual rate of falls=30% - do you mean that 30% of those aged 65 and older will fall every year? Please reconsider how you have phrased this as falls rates are not usually expressed as a % like this.

2. Materials and Methods - this section would benefit from subheadings (and potentially sub-subheadings) to separate intervention conditions and outcome measure details. 

Where and how were participants recruited? From the community? Through the hospital?

Exclusion criterion 1: can you clarify how you defined and assessed "lack of regular exercise" and "not received exercise instruction"? E.g. Asked each participant how many minutes of exercise they do a week, and if they report less than X minutes then they are eligible. Do you also include incidental exercise? 

Line 98: "Eligible participants were randomly assigned to ONE OF three experimental groups OR a control group."

Please add more details for the experimental conditions. Your abstract stated each WBV session was 5 minutes, this also needs to be stated in the methods section. Were the participants given any specific instructions about how to stand? How were the WBV and heat therapy scheduled? WBV first or heat first? Was this order rotated and randomised across participants? Please also state what the control group was instructed to do: nothing? usual lifestyle? no instructions?

Line 110: was familiarisation conducted before every assessment or only the first one?

Line 126: reference to figure two for LOS calculations seems like an error? Figure 2 shows a person using the heating pad.

3. Results - this section is missing the raw data and analysis of the changes across the 3 time points. Your figures are helpful and illustrative, but we need the actual numbers reported for changes across time and between groups like you did in Table 1 for all the pre-test outcomes for between and within group comparisons. Report everything you said you would in the last paragraph of the methods section. I also suggest adding a summary paragraph at the start of the results section to briefly go over the outcome measures that saw significant changes, and then go into the details of the changes for each one.

Figures 4-6: please label the time points so that readers can see easily see it is pre-, post- and 6 months post. Please also consider changing the labels for each line to the experimental condition rather than just numbers, this will be easier for readers.

4. Discussion - needs to be better organised and missing discussion of follow-up results.

Line 200: the use of the term "functional" suggests to me that you consider the sit-and-reach test to be a functional task, but this is not the case.

Line 203-204: "The WBV plus heat therapy group had significantly more improvement than the other three groups. This finding is consistent with previous studies." - I thought there were no previous studies investigating this combination? If the studies producing similar results only used one without the other, then your second statement would not be accurate. Please be careful about how you are phrasing your discussion points.

Line 228-229: please add what the temperature of the muscles and ligaments are

Last paragraph in results: you start by stating that significant improvements were observed for balance in the WBV and WBV+heat groups, then state the exact opposite in lines 240-241. Figure 6 seems to show that there were significant improvements. This is a major conflict in your manuscript, please resolve this during your amendments and ensure you are CLEARLY reporting your findings.

There are no discussions about time point 3 (6 months following the 3 months of intervention). It appears that flexibility improvements were maintained for WBV and even continued to improve for WBV, strength and balance went down but was still higher in the WBV+heat group. These are all worthy of discussion in relation to how they might be reflective of long term benefits of WBV and WBV+heat.

Author Response

This is a very interesting study examining the effects of whole-body vibration and heat therapy for physical function. The design and outcome measures appear appropriate, but there are key parts of the results that have not been adequately reported. The discussion will also benefit from additional content after the authors have addressed the missing elements in the results section.

Abstract

- Please add a brief sentence for what the heat therapy involved, just like how you described the WBV condition.

Response: Thank you for your insightful comment. We have added the sentence, The feasibility and applicability of thermotherapy are increasingly being demonstrated by applications and clinical trials, in the abstract as suggested.

- Please clarify what pre-test and post-test are in reference to: each exercise session? Or the whole 3-month block?

Response: Thank you for the question. A comparison of training before and after three months are in a training course.

- Keywords: you could add more keywords to make your paper more searchable, e.g. exercise, strength training, flexibility training. Consider using MeSH terms.

Response: Thank you for the suggestion. Balance training and flexibility training have been added in the keyword list.

  1. Introduction

Line 38-39: annual rate of falls=30% - do you mean that 30% of those aged 65 and older will fall every year? Please reconsider how you have phrased this as falls rates are not usually expressed as a % like this.

Response: Thank you for the insightful comment. The sentence whereas the annual rate among those over 80 years of age exceeds 50% has been added in the text.

  1. Materials and Methods - this section would benefit from subheadings (and potentially sub-subheadings) to separate intervention conditions and outcome measure details. 

Response: Thank you for the suggestion. The suggested subheadings have been added in Line 107 and 122.

Where and how were participants recruited? From the community? Through the hospital?

Response: Thank you for the questions. The participants were recruited from the community and through the hospital.

Exclusion criterion 1: can you clarify how you defined and assessed "lack of regular exercise" and "not received exercise instruction"? E.g. Asked each participant how many minutes of exercise they do a week, and if they report less than X minutes then they are eligible. Do you also include incidental exercise? 

Response: Thank you for the insightful comment. The information has been added in Line 100 to 102.

Line 98: "Eligible participants were randomly assigned to ONE OF three experimental groups OR a control group."

Please add more details for the experimental conditions. Your abstract stated each WBV session was 5 minutes, this also needs to be stated in the methods section. Were the participants given any specific instructions about how to stand? How were the WBV and heat therapy scheduled? WBV first or heat first? Was this order rotated and randomised across participants? Please also state what the control group was instructed to do: nothing? usual lifestyle? no instructions?

Response: Thank you for the insightful comment. The information has been added in Line 112 to 113, 115, and 118-119.

Line 110: was familiarisation conducted before every assessment or only the first one?

Response: Thank you for the questions. Only the first one familiarization was conducted. Testing took place before the experiment, three months after training began, and six months after training.

Line 126: reference to figure two for LOS calculations seems like an error? Figure 2 shows a person using the heating pad.

Response: Thank you for the insightful comments. The corrections have been made in LINE 138-140.

  1. Results - this section is missing the raw data and analysis of the changes across the 3 time points. Your figures are helpful and illustrative, but we need the actual numbers reported for changes across time and between groups like you did in Table 1 for all the pre-test outcomes for between and within group comparisons. Report everything you said you would in the last paragraph of the methods section. I also suggest adding a summary paragraph at the start of the results section to briefly go over the outcome measures that saw significant changes, and then go into the details of the changes for each one.

Figures 4-6: please label the time points so that readers can see easily see it is pre-, post- and 6 months post. Please also consider changing the labels for each line to the experimental condition rather than just numbers, this will be easier for readers.

Response: Thank you for the insightful comment. The information has been made to all the necessary figures as suggested.

  1. Discussion - needs to be better organised and missing discussion of follow-up results.

Line 200: the use of the term "functional" suggests to me that you consider the sit-and-reach test to be a functional task, but this is not the case.

Response: Thank you for the insightful comment. The term has been revised in LINE 225.

Line 203-204: "The WBV plus heat therapy group had significantly more improvement than the other three groups. This finding is consistent with previous studies." - I thought there were no previous studies investigating this combination? If the studies producing similar results only used one without the other, then your second statement would not be accurate. Please be careful about how you are phrasing your discussion points.

Response: Thank you for the suggestion. The corrections have been made in LINE 225-234.

Line 228-229: please add what the temperature of the muscles and ligaments are

Response: Thank you for the suggestion. The information has been added in LINE 250.

Last paragraph in results: you start by stating that significant improvements were observed for balance in the WBV and WBV+heat groups, then state the exact opposite in lines 240-241. Figure 6 seems to show that there were significant improvements. This is a major conflict in your manuscript, please resolve this during your amendments and ensure you are CLEARLY reporting your findings.

Response: Thank you for the kind suggestion. The paragraph has been revised in LINE 267-273.

There are no discussions about time point 3 (6 months following the 3 months of intervention). It appears that flexibility improvements were maintained for WBV and even continued to improve for WBV, strength and balance went down but was still higher in the WBV+heat group. These are all worthy of discussion in relation to how they might be reflective of long term benefits of WBV and WBV+heat.

Response: Thank you for the insightful comment. It has been shown in previous studies that if training stops for more than three to eight weeks, the arterial oxygen difference decreases. The rapid and progressive decrease in oxidase activity leads to a decrease in mitochondrial ATP production. These changes are associated with the decrease in VO2max observed during prolonged cessation of training. Non-athletes with short-term training usually return to their baseline values after a short period of time without exercise (Mujika & Padilla, 2001). It is therefore crucial to maintain long-term exercise training.

Reviewer 2 Report

Whole-body vibration (WBV) is a novel exercise training measure that promotes the muscle strength, flexibility, and balance abilities of elderly groups. However, no studies have examined the interactions between the pre-exercise and post-exercise application of heat therapy.

The authors investigate the effects of WBV and heat therapy on the muscle strength, flexibility, and balance abilities of elderly groups. Eighty middle-age and elderly participants with no regular exercise habits were enrolled in this study. They were randomly assigned to a WBV group, a WBV plus heat therapy group, a heat therapy alone group, and a control group. The WBV groups underwent 5-min, fixed-amplitude (4 mm), thrice-weekly WBV training sessions for 3 consecutive months on a WBV training machine. Participants’ balance was measured using the limits of stability (LOS) test on a balance system.

The authors tested the pretest and posttest knee extensor and flexor strength using an isokinetic lower extremity dynamometer.

They measured the pretest and posttest flexibility changes ured using the sit-and-reach test. Significantly larger pretest and posttest differences in flexibility and muscle strength were observed in the WBV and WBV plus heat therapy groups. The addition of heat therapy to WBV resulted in the largest flexibility improvements.

The study is interesting.

I propose some minor comments with a pure academic spirit:

1.     Add the conclusions in the abstract

2.     A clear purpose is lacking. Please add it at the end of the Intro or in separated paragraph.

3.     Please add the labels with data in figures 4-6.

4.     Please insert the limitations of the study in the discussion.

Author Response

Response: Thank you for the insightful comments. The corrections have been made in LINE 100-119, 190-192, 196-200, 208-211, 218-235, 266-273, and 286-288.

Reviewer 3 Report

Authors have studied the effect of WBV and heat therapy on older adults' muscle strength and balance abilities. The paper requires detailed statistical analysis of the data to prove their claims.

Introduction: 

Line 48: Briefly overview the existing practices and limitations of the current practices and explain the research gap.

At the end of the introduction section, describe how You have validated your solution (description of case studies and/or benchmarks. Why these benchmarks/case studies are important and interesting. What is the motivation behind selecting these benchmarks). Also describe briefly what are the significance and impact of your solution.

Materials and Methods

Please describe the methodology with some useful figures (block diagrams) such that each figure must be explained properly in the text. 

Line 126: Figure 2 is different, as explained in the text.

Results

Line 169: In figure 3, group 2 should be linked to the vibration and heat groups.

Figure 4: time should be mentioned as months.

Results should be tabulated for all parameters with their p-values. Only figures are not sufficient.

Figure 5: Muscle strength for different graphs. Improvement is not established without statistical analysis. Moreover, why did the muscle strength increase after two months and decrease suddenly after three months? This phenomenon is strange and not compatible with figure 4. For groups, 1,2, and 3 difference in final muscle strength after three months is not proven statistically. Moreover, here group 1 is better than group 2; why?

 Line 190: Data is unavailable in support of the claim significance and not significance.

Discussion

Line 203: Does WBV and heat therapy group outperform the WBV only group in all parameters under the study?

Line 218: This claim also needs statistical analysis on the groups.

Comparing these results with other similar strategies published previously may be a good idea.

Moreover, the effect of the duration of exercise and other parameters related to WBV and heat therapy may also be discovered.

Conclusions

The conclusion is very short.

References

More recent references may be added.

Author Response

Authors have studied the effect of WBV and heat therapy on older adults' muscle strength and balance abilities. The paper requires detailed statistical analysis of the data to prove their claims.

Response: Thank you for the insightful comments. The corrections have been made in LINE 100-119, 190-192, 196-200, 208-211, 218-235, 266-273, and 286-288.

Introduction: 

Line 48: Briefly overview the existing practices and limitations of the current practices and explain the research gap.

Response: Thank you for the insightful comments. The corrections have been made in LINE 51-52.

At the end of the introduction section, describe how You have validated your solution (description of case studies and/or benchmarks. Why these benchmarks/case studies are important and interesting. What is the motivation behind selecting these benchmarks). Also describe briefly what are the significance and impact of your solution.

Response: Thank you for the insightful comments. The corrections have been made in LINE 51-52.

Materials and Methods

Please describe the methodology with some useful figures (block diagrams) such that each figure must be explained properly in the text. 

Line 126: Figure 2 is different, as explained in the text.

Response: Thank you for the insightful comments. The corrections have been made in LINE 138-140.

Results

Line 169: In figure 3, group 2 should be linked to the vibration and heat groups.

Response: Thank you for the insightful comments. The corrections have been made in LINE 181.

Figure 4: time should be mentioned as months.

Response: Thank you for the insightful comments. The corrections have been made in LINE 194-197.

Results should be tabulated for all parameters with their p-values. Only figures are not sufficient.

Figure 5: Muscle strength for different graphs. Improvement is not established without statistical analysis. Moreover, why did the muscle strength increase after two months and decrease suddenly after three months? This phenomenon is strange and not compatible with figure 4. For groups, 1,2, and 3 difference in final muscle strength after three months is not proven statistically. Moreover, here group 1 is better than group 2; why?

Response: Thank you for the insightful comments. The corrections have been made in LINE 207-211.

Line 190: Data is unavailable in support of the claim significance and not significance.

Response: Thank you for the insightful comments. The corrections have been made in LINE 218-221.

Discussion

Line 203: Does WBV and heat therapy group outperform the WBV only group in all parameters under the study?

Line 218: This claim also needs statistical analysis on the groups.

Comparing these results with other similar strategies published previously may be a good idea.

Moreover, the effect of the duration of exercise and other parameters related to WBV and heat therapy may also be discovered.

Response: Thank you for the insightful comments. The corrections have been made in LINE 225-234, and 266-273.

Conclusions

The conclusion is very short.

Response: Thank you for the insightful comments. The corrections have been made in LINE 285-288.

References

More recent references may be added.

Response: Thank you for the insightful comments. The references have been added in the list.

Round 2

Reviewer 1 Report

The authors have made some amendments to the manuscript based on my initial feedback, but a number of major issues remain in the reporting of the study findings. 

Abstract

My previous comment about adding a brief sentence for heat therapy was to ask you to add details about the schedule and intensity of the heat therapy, like your details for the WBV in terms of time and amplitude etc.

Thank you for answering my questions, but it is not sufficient to simply address them in your responses, they need to be shown in the manuscript as well. If I have asked a question, it is because your manuscript was not clear enough, therefore I suggest the authors to include your clarifications in your manuscript as well. 

Introduction

Line 42: your added sentence does not address my query, my feedback was that how you have written the sentence isn't specific enough. My suggestion is to rephrase your reporting of falls rates to "Up to X% of older adults aged XX year and over have reported at least one fall during their medical examinations..." or something similar.

Methods: I'm satisfied with the amendments

Results

Figure legends haven't been edited. The authors added numbers to each group in the flowchart, but this still requires readers to scroll back and forth to match the group number to the group condition. I'm suggesting that you edit the legends IN THE FIGURE so that readers can make sense of the graph at first glance, without having to check what each group number or time point number actually means. It looks like your figures were generated in SPSS, you can edit figure labels and legends in SPSS.

The newly added tables are very helpful, but please check them for formatting issues: they are not numbered and there are spelling mistakes. E.g. Table on page 6, column headings are poorly spaced, month is misspelt as moth, label for p-value column is not properly labelled. 

All your tables only report differences between time 1 and time 2. Time 3 also needs to be considered. 

Line 191-192: additional phrase has made this a very long run on sentence, please rewrite this whole sentence for clarity.

Discussion

Line 230-231: "WBV plus heat therapy groups significant higher than WBV group." higher in what measures? This newly added section is helpful, but you need to make sure it blends in well with what you have already written. I suggest editing to get rid of repetitive sentences and try to express your message in a more efficient way.

Newly added section in lines 267-273 does not address my issue about conflict in your last paragraph. Your last paragraph starts with "The WBV and WBV plus heat therapy groups significantly improved with balance abilities." (line 275-276) and then later in the same paragraph it says "No significant improvements in balance abilities were observed in the WBV and WBV plus heat therapy groups," (lines 279-280) - this is the problem I'm talking about, these two sentences are contradicting each other!

This newly added section (lines 267 -273) also doesn't address my issue about the lack of comparisons and discussions for time 3. If you're not going to compare the changes from time 1 and time 2 to time 3, then what is the point of including this final time point? 

Author Response

Please see attacehd

Reviewer 3 Report

Authors have revised the paper and most of my concerns are answered in the revised version. There are few modifications required in the revised version as follows,

Line 48: Briefly overview the existing practices and limitations of the current practices and explain the research gap.

Response: Thank you for the insightful comments. The corrections have been made in LINE 51-52.

New Comment:  Response is not enough. Adding a table to highlight the limitations and performance of existing methods should be included.

New Comment:  Figures 4 and 5, Add the standard deviation with the mean value on the graph.

Figure 5: Muscle strength for different graphs. Improvement is not established without statistical analysis. Moreover, why did the muscle strength increase after two months and decrease suddenly after three months? This phenomenon is strange and not compatible with figure 4. For groups, 1,2, and 3 difference in final muscle strength after three months is not proven statistically. Moreover, here group 1 is better than group 2; why?

Response: Thank you for the insightful comments. The corrections have been made in LINE 207-211.

New Comment:  The comment “why did the muscle strength increase after two months and decrease suddenly after three months? This phenomenon is strange and not compatible with figure 4.” is not answered in the revised version.
